# The Establishment of Fibrolytic Bacteria in the Foal Gastrointestinal Tract Is Related to the Occurrence of Coprophagy by Foals

**DOI:** 10.3390/ani13172718

**Published:** 2023-08-26

**Authors:** Morgan Pyles, Miranda Agbana, Susan Hayes, Michael Flythe, Laurie Lawrence

**Affiliations:** 1Department of Animal and Food Sciences, University of Kentucky, Lexington, KY 40546, USA; pyles024@crk.umn.edu (M.P.); michael.flythe@usda.gov (M.F.); 2Forage-Animal Production Research Unit, Agricultural Research Service, United States Department of Agriculture, Lexington, KY 40546, USA

**Keywords:** cellulolytic bacteria, coprophagy, fiber, foals, microbial colonization

## Abstract

**Simple Summary:**

Horses rely on specific functional groups in the microbial community in their gastrointestinal tract to digest fiber-rich feedstuffs such as hay and pasture. The microbial community is not fully functional at birth but changes and develops with age. The factors directing the development of the microbial community in the foal are not well understood. The consumption of maternal feces, coprophagy, is part of the normal behavior in the foal and may contribute to microbial colonization. To investigate coprophagy in foals, mares were fed an indigestible marker that was excreted in their feces. When the foals consumed maternal feces, the marker was detected in the foal’s feces. The results suggest that coprophagy is important for the establishment of fibrolytic bacteria in the foals’ gastrointestinal tract. Maternal feces are rich in fiber and live microbes, providing both a prebiotic and a probiotic to the foal.

**Abstract:**

The consumption of maternal feces (coprophagy) is commonly observed in healthy foals and is a proposed contributor to microbial colonization of the foal’s gastrointestinal tract (GIT). This study investigated the role of coprophagy in the establishment of fibrolytic bacteria in the foal GIT. Nine thoroughbred mares were dosed with chromic oxide, an indigestible marker, as a method to detect the occurrence of coprophagy by their foals. Foal fecal samples were collected from 12 h to 21 d after birth to measure chromic oxide and neutral detergent fiber (NDF) and to enumerate cellulolytic bacteria using culture-based techniques. Milk yield was estimated at 7 and 14 d postpartum. Coprophagy was detected as early as 3 d after birth and detected in all foals by 7 d of age. There were strong relationships between coprophagy and cellulolytic bacteria and NDF in foal feces at 7 d of age (r = 0.9703 and r = 0.7878, respectively; *p* < 0.05). Fecal NDF and chromic oxide concentrations were negatively related to milk yield (r = –0.8144 and r = –0.6966, respectively; *p* < 0.05), suggesting milk availability affected the incidence of coprophagy. Based on the relationships identified, maternal feces are an important source of fiber and live microbes for the foal, contributing to the development of the microbial community.

## 1. Introduction

Coprophagy has been noted as part of normal foal behavior [1]. There have been several proposed roles of coprophagy in foal health, including serving as a source of inoculum for the microbial community, aiding in the development of food preferences, and as a source of some vitamins or other nutrients [2]. Previous research demonstrated that, when given an option, foals show a strong preference for their dam’s feces over the feces of other adult mares [3]. Additionally, there is evidence that the similarity among fecal microbial communities is greater between foals and their dams than between other foals [4]. In an investigation of the microbial richness in foal feces and occurrence of coprophagy in the first 4 d after birth, the first observed bouts of coprophagy occurred at 2 d of age, yet microbial DNA was already detected in foal feces within the first 24 h after birth [5]. The microbial community in the foal GIT changes with age; amylolytic bacteria and lactobacilli are detected soon after birth, whereas cellulolytic bacteria are not detected until 4 to 7 d after birth [6,7]. These observations suggest that coprophagy is not essential for the initiation of microbial colonization of the gastrointestinal tract (GIT); however, coprophagy may influence the colonization of specific functional groups that arrive after the pioneer species. We hypothesized that a relationship exists between coprophagy and the establishment of the cellulolytic bacteria in the foal’s GIT. 

During the neonatal period, foals rely primarily on mare milk for their nutrient needs. However, foals begin the ingestion of solid food within the first few days after birth and the time spent foraging increases with age in foals [1,2,8]. Increasing foraging activity subsequently increases the ingestion of structural carbohydrates. Horses do not produce enzymes capable of digesting structural carbohydrates; instead, structural carbohydrates are utilized as substrate for the microbial community in the GIT. Thus, for fibrolytic organisms to colonize the foal’s GIT, structural carbohydrates must be present as substrate. We hypothesized that a relationship exists between the establishment of fibrolytic bacteria and the presence of structural carbohydrates in the foal’s GIT. 

In this study, chromic oxide, an indigestible marker, was fed to mares or orally dosed to foals. The objectives were three-fold. First, we used chromic oxide fed to mares to (1) evaluate the relationship between coprophagy, fecal fiber, and cellulolytic bacteria in foal feces, and (2) determine the influence of mare milk yield on the onset and occurrence of coprophagy in foals and the number of cellulolytic bacteria in the foal’s feces. For the third objective, foals were orally dosed with chromic oxide to assess gastrointestinal transit time.

## 2. Materials and Methods

### 2.1. Animals and Management 

All procedures used in this study were approved by the University of Kentucky’s Institutional Care and Use Committee. Twelve thoroughbred mares and their foals and five additional foals were enrolled in this project. Prior to parturition, pregnant mares were housed in outdoor paddocks during the day and individual box stalls (3.5 × 3.5 m) at night until parturition. For 1 to 7 days after parturition, mares and foals remained in individual stalls at night and individual pens during the day. Mares and foals were comingled with other mare/foal pairs by approximately 1 week postpartum in small groups (2 to 4 pairs per paddock). An additional five foals were used to estimate gastrointestinal transit time. Forage (mixed grass and alfalfa hay and cool-season grass pasture) and water were available to mares and foals ad libitum throughout the study. All mares were fed a commercial concentrate throughout the study (Table 1; Original 14, McCauley Bros Inc., Versailles, KY, USA). Before parturition, mares were fed concentrate twice daily, which increased to three meals per day postpartum. The twelve mares assigned to receive chromic oxide were adapted to concentrate feeding in individual nosebags prior to parturition and were fed concentrate in nosebags throughout the study. 

### 2.2. Detecting Coprophagy in Foals

Chromic oxide is considered an indigestible marker; thus, when fed, it should appear in mare feces. Subsequently, when the foals consume maternal feces, the marker will appear in the foal’s feces. The chromic oxide was fed to twelve mares using soft, pliable treats (Figure 1a, Dimples Horse Treats, Union, KY, USA). The treats were a proprietary blend of molasses, dry molasses, ground corn, oats, soybean meal, ground flax, wheat flour, beet pulp, and wheat bran. The chemical composition of the treats can be found in Table 1.

The mares were adapted to the treats by first breaking up the treat into small pieces and mixing with the concentrate meal fed in nosebags. When the mares consumed the entire meal and all the treat material, whole treats were added to their nosebag. Once adapted to consuming the treats, chromic oxide (2 g) was mixed with molasses (1.75 g) to form a paste and placed in the center of a treat, which was then formed into a ball (Figure 1b). The chromic oxide-containing treat was placed in the nosebag with concentrate meals. Of the 12 mares in the study, 3 refused the treats; thus. 9 mares were dosed with chromic oxide for detection of coprophagy in their foals. Each mare received 10 g of chromic oxide divided equally between concentrate meals beginning at 333 d of gestation to ensure a steady state of chromic oxide excretion in feces was attained by parturition. The chromic oxide dosing continued until 21 d postpartum.

The 10 g/day dosing was selected based on previous research using chromic oxide to determine the digestibility of nutrients in adult horses [10]. The previous study used a dose of 6.84 g chromic oxide/d and reported a minimum fecal recovery of 72%. The average daily fecal output in the study by Fowler et al. was 3.5 kg (DM) in adult horses. The mares in the current study had an increased dry matter (DM) intake compared to the previous study and were expected to have an estimated daily fecal output of 4.5 kg (DM)/d. Assuming a similar recovery of chromic oxide in mare feces, the dose was increased to 10 g/d to ensure an adequate concentration of chromic oxide in mare feces to detect the marker in foal feces after the consumption of maternal feces. When dosed with 10 g of chromic oxide/d, we expected the mare feces to contain an average of 1.6 g marker/kg (DM) or 1600 ppm. Fecal samples were collected from six of the nine mares fed chromic oxide to provide an estimate of the chromic oxide and neutral detergent fiber (NDF) available to the foals via coprophagy.

### 2.3. Fecal Sample Collection

Fecal samples were collected from the foals of mares fed chromic oxide at the following time points: 12 h, 2, 3, 5, 7, 10, 14, and 21 d after birth. Fresh fecal samples were manually collected during, or immediately after, defecation into sterile specimen cups using sterile gloves to avoid contamination. The time of collection was recorded for each sample collected. A subsample was removed for bacterial enumerations in samples collected at 12 h, 3, 7, 14, and 21 d and the remaining feces were frozen for subsequent analyses of DM, NDF and chromic oxide. Fecal samples were also collected from the five thoroughbred foals of mares not fed chromic oxide. These foals were dosed orally with chromic oxide to evaluate the time between consumption of chromic oxide by foals and the appearance of chromic oxide in foal feces. The dosage administered orally at 4 or 6 d of age (average 4.4 d of age) was 0.4 g of chromic oxide mixed in 5 mL of water. The foals were randomly dosed at either 0700 or 1900 h. All feces excreted between 0600 and 2000 h were collected from the foals for the subsequent 4 d post-dosing. Because feces were not collected over night, the two dosing times were selected to capture the excretion pattern of the marker and to account for potential diurnal effects on excretion. The time of each collection was recorded for all feces. Fecal samples were frozen and subsequently analyzed for dry matter and chromic oxide as described below. 

### 2.4. Bacterial Enumerations in Foal Feces

Fecal samples collected at 12 h, 3, 7, 14, and 21 d were used to enumerate cellulolytic bacteria in foals from mares fed chromic oxide. Immediately following collection, fresh feces were mixed by hand and then an approximately 1 g subsample was placed in a sterile, pre-warmed Hungate tube. Air was removed from the tube via displacement with CO_2_ using a sterile tuberculin needle. Fecal samples were transported to the lab in a warm container (37 °C) within 2 h of collection. Sterile phosphate-buffered saline (PBS) was used to dilute the sample 1:10 after the actual sample weight was determined. A dilution series was then established using PBS as the diluent. Dilutions were used to inoculate media for the enumeration of cellulolytic bacteria as described previously [11]. Briefly, cellulolytic bacteria were enumerated using an anaerobic medium with a strip of cellulose filter paper (Whatman #1; 4 g/L media) as the sole substrate. The dissolution of cellulose after 10 d of incubation (37 °C) was used as an indicator of bacterial growth. The highest dilution exhibiting bacterial growth was recorded. 

### 2.5. Analysis of Fecal Neutral Detergent Fiber and Chromic Oxide

Foal feces at 12 h, 2, 3, 5, 7, 10, 14, and 21 d were dried to a steady state in a 55 °C forced-air oven, then ground in a Cyclotec Sample Mill (Foss, Hillerod, Denmark). Dried ground samples were used to determine NDF and chromic oxide content. Fecal NDF was determined using an Ankom 200 Fiber Analyzer (Ankom Technology, Macedon, NY, USA) following the manufacturer’s instructions. To analyze the chromic oxide concentration in fecal samples, a 0.5 g sample of dried, ground feces was placed in a quartz crucible and ashed at 600 °C overnight. The samples were then digested using potassium bromate and manganese sulfate. Digested samples were placed in specimen cups and then diluted to 100 mL with a calcium chloride solution and allowed to stand overnight. The chromium in the solution was analyzed using atomic absorption at a wavelength of 357.87 nm (AAnalyst 200, PerkinElmer Inc., Waltham, MA, USA) and the concentration was calculated using a six-point standard curve (1–7 ppm).

### 2.6. Estimating Daily Milk Yield 

To address the second objective of the study, daily milk yield in mares fed chromic oxide was estimated at the end of week 1 postpartum (days 6 and 7; n = 9) and at the end of week 2 postpartum (days 14 and 15 d; n = 8). On each day, mare and foal pairs were brought into box stalls and the foal was allowed to nurse until satisfied. The foal was then muzzled for 2 h, after which the muzzle was removed, and the foal allowed to nurse one side of the udder. While the foal was nursing, the opposite side of the udder was milked using a handheld mare milker (Udderly EZ, EZ Animal Products, IA, USA). Milking ended when nursing by the foal ceased. The muzzle was placed on the foal for an additional 2 h, followed by the same milking procedure on the same side of the udder. The following day, the entire 4 h milking procedure was repeated using the opposite side of the udder. The order in which each side of the udder was milked was randomized with each mare.

The volume of milk collected (L) at each milking was recorded and used to calculate daily milk yield in L/d and on a BW basis using the following equations:$$\mathrm{Milk}\;\mathrm{yield},\;L/d=\frac{M}{I}\times \frac{24\;h}{d}\times 2$$
$$\mathrm{Milk}\;\mathrm{yield},\;\%\;\mathrm{of}\;\mathrm{BW}=\left(\frac{\frac{M}{I}\times \frac{24\;h}{d}\times 2}{BW}\right)\times 100$$
where M represents the volume of milk (L) collected from one side of the udder during one milking, I represents the interval between the milkings (in h), and BW represents mare body weight (kg).

### 2.7. Statistical Analyses

All statistical analyses were carried out using SAS software version 9.4 (SAS Institute Inc., Cary, NC, USA). Changes with age in the concentration of chromic oxide and cellulolytic bacteria in foal feces were analyzed using mixed model ANOVA with repeated measures. Data were tested for normality and transformed when necessary prior to analysis. Means were separated using an LSD test and Kenward–Rogers approximation was used to estimate degrees of freedom. The covariance structure of auto-regressive order 1 was used in the repeated statement, use of the heterogeneous version was determined based on Akaike’s and Bayesian Information Criteria. Enumeration data were log10-transformed for normalization before statistical analysis. 

The relationships among variables were evaluated using regression analysis and Pearson’s correlation coefficient (Proc Corr). Significance was determined when *p* < 0.05 and a trend was considered when *p* < 0.10.

## 3. Results and Discussion

### 3.1. Detecting Coprophagy in Foals

To evaluate the occurrence of coprophagy in foals, nine mares were fed chromic oxide-containing treats with their concentrate meals to detect chromic oxide in foal feces after consumption of maternal feces. The intention was to collect fecal samples from all foals at 12 h, 2, 3, 5, 7, 10, 14 and 21 d after birth. However, due to infrequent defecation by foals, fecal samples were not collected from all foals at every time point. The average concentration of chromic oxide in maternal feces was 1143.65 ppm with a range of 849.30 to 1417.78 ppm (n = 6). The concentration of chromic oxide in the spot samples from the mares was slightly lower than we had anticipated (1600 ppm) but was adequate in allowing for the detection of chromic oxide in their foals’ feces. 

Chromic oxide was detected in the feces of all nine foals from mares that consumed the chromic oxide (Figure 2). One foal had detectable chromic oxide at 3 d of age, but by 7 d, all foals had detectable chromic oxide in their feces. The highest concentration of chromic oxide for the majority of the foals occurred between 10 and 14 d of age. The concentration of chromic oxide varied in individual foal fecal samples, ranging from 0 ppm to 540.13 ppm across all time points, which could be reflective of the amount of coprophagy taking place.

Finding detectable amounts of chromic oxide in feces of all foals by 7 d of age as an indicator of coprophagy is novel, but these results do not provide information on when the coprophagy took place. The rate of digesta passage in foals is currently unknown. Thus, five foals from mares not fed chromic oxide were given chromic oxide orally at 4 to 5 days of age. After oral dosing with chromic oxide, fecal samples were collected from foals from 2 to 102 h post-dosing. The time from dosing to the first chromic oxide detected in foal feces was an average of 33 h (range 17 to 54 h). The majority of the marker was excreted between 25 and 96 h post dosing (Table 2). These results indicate that 1 to 2 days elapsed between the consumption of chromic oxide in mare feces and the appearance of marker in foal feces. Taken together, these data indicate that most foals begin to consume maternal feces between 2 and 5 d of age. The findings from this study are in agreement with previous observational data in regard to the onset of coprophagy in foals [12]. 

### 3.2. Coprophagy and Cellulolytic Bacteria

We hypothesized that a relationship would exist between the appearance of cellulolytic bacteria in foal feces and the occurrence of coprophagy. When the changes with age in cellulolytic bacteria were evaluated, no cellulolytic bacteria were detected in foal feces at 12 h or 3 d of age, but by 7 d of age, five of the nine foals had detectable cellulolytic bacteria. Cellulolytic bacteria were detected in all foals by 14 d of age and the number of bacteria increased from 14 to 21 days of age (Figure 3). These results are consistent with previous studies demonstrating that cellulolytic bacteria are not a pioneer species in the foal GIT but begin to appear in foal feces within the first 2 weeks of life [6].

An objective of the current study was to investigate the establishment of cellulolytic bacteria in relation to the consumption of maternal feces by foals. The relationship between chromic oxide concentration and cellulolytic bacteria in foal feces at 7 d of age was very strong (r = 0.9703, *p* < 0.0001; Table 3). It has been suggested that coprophagy may play a role in foal health by serving as a source of inoculum or probiotic for the microbial community [2]. Maternal feces contain viable cellulolytic bacteria [13] and the highly correlated relationship between coprophagy and cellulolytic bacteria in this study supports the theory that maternal feces may serve as a natural probiotic to the foal. Probiotics are described as live microorganisms that confer a health benefit on the host when administered in adequate amounts [14]. Cellulolytic bacteria are obligate anaerobes and will benefit a foal as the need for nutrients from fiber-rich feedstuffs increases. Because they do not survive well in an aerobic environment, fresh maternal feces provide a source of live cellulolytic bacteria readily available to inoculate the foal’s GIT. It should be noted that previous studies have identified viable bacteria in foal feces as early as 12 h post foaling, including amylolytic bacteria, lactobacilli, and lactate-utilizing bacteria [7,15]. Observations here and in other studies suggest that the early bacteria appear in the GIT before foals have started to consume maternal feces. Consequently, the consumption of maternal feces may not be the route of inoculation for the pioneer species that are abundant within the first few days after birth but may be an important source of fiber digesting organisms that are established later. 

### 3.3. Neutral Detergent Fiber in Foal Feces

In addition to evaluating the occurrence of coprophagy in foals, another objective was to evaluate the relationships among the foals’ diet (including fiber sources and mare milk), coprophagy, and microbial colonization. The NDF content of foal feces was analyzed as an indicator of available substrate for cellulolytic bacteria. During the neonatal period, foals rely on maternal milk as their primary nutrient source. Several studies have demonstrated that as foals age, their time spent nursing decreases with a concomitant increase in the time spent foraging [1,8,16]. The studies by Duncan et al. [16] and Crowell-Davis et al. [8] evaluated the time budget of foals, whereas Bolzan et al. [1] estimated both grazing time and estimated bite mass and dry matter intake in foals. 

In the current study, the concentration of NDF in foal feces changed over time (*p* < 0.0001; Figure 4). Following a similar trend as the number of cellulolytic bacteria, there was minimal fiber detected in foal feces from 12 h to 3 d of age; then, the NDF content in foal feces increased progressively through 21 d of age. The concentration of fecal NDF between foals varied. For example, at 7 d of age, NDF ranged from 6.63% to 36.28% (DM basis). These data demonstrate that foals appear to begin consuming fiber within the first week of life and the amount of fiber consumed continues increasing within the first 3 weeks after birth. The study by Bolzan et al. [1] detected an increase in the time foals spent grazing from 5–8 days of age to 20–22 days of age (0.5 to 13 min, respectively). When the time spent grazing increases, it would be expected that the amount of fiber detected in foal feces also increases.

As foals begin to consume more fiber, there are several potential sources of fiber available to foals. The mares were housed in outdoor paddocks during the day with cool-season grasses available to both the mares and foals in addition to ad libitum grass and alfalfa mixed hay available in the stalls and paddocks. An additional source of fiber available to the foals could be maternal feces. In this study, fecal samples were collected from six of the nine mares dosed with chromic oxide and the NDF content was analyzed to give an estimate of the fiber content available to the foals through coprophagy. The mare feces contained an average of 54.2% NDF. 

All the foals in this study had consumed maternal feces during the first 3 wk of life, evidenced by the detection of chromic oxide in their feces. The concentration of fiber in foal feces was positively related with the concentration of chromic oxide from day 7 through 21 d of age (Table 3). The fiber in maternal feces may provide substrate for cellulolytic bacteria in the foal’s GIT via coprophagy; there was a positive relationship between foal fecal NDF and the number of cellulolytic bacteria at 7 d of age (r = 0.8607, *p* = 0.029), but the relationship was not significant at 14 or 21 d of age (*p* > 0.05). The continued relationship between coprophagy and fiber indicates that maternal feces provide an important source of fiber for young foals.

Maternal feces also contain a diverse microbial community [17,18] that may provide a source of inoculum for the foal’s developing microbiome, such as cellulolytic bacteria. Maternal feces, therefore, may serve in two roles, as a probiotic in providing cellulolytic bacteria to the foals, and as a natural prebiotic for the foal, by providing fiber. To be classified as a prebiotic, the substance should be resistant to acidic pH, not hydrolyzed by mammalian enzymes or absorbed in the GIT, and can be fermented by intestinal microbes [19]. In addition to serving as a prebiotic for fiber-digesting bacteria, the highly correlated relationship between coprophagy and cellulolytic bacteria at 7 d may be due to both the prebiotic and probiotic effect by supplying a source of live cellulolytic bacteria that may aid in inoculating the foal GIT and supplying the fiber needed as a substrate for the proliferation of these bacteria. 

### 3.4. Relationships among Foal Fecal Fiber, Coprophagy, and Milk Yield

Milk yield (L/d) was used to evaluate relationships among coprophagy, fecal bacteria, and fiber. Daily milk yield was estimated at two time points, 7 d and 14 d postpartum. At 7 d postpartum, daily milk yield was 12.33 ± 0.99 L/d with a range of 8.22 to 17.39 L/d and then increased to 15.62 ± 1.06 L/d with a range of 11.1 to 19.92 L/d (*p* = 0.0386; Table 4). Peak milk production in mares occurs between 1 and 2 mo postpartum and can reach 3% of the mares BW [20,21]. By 2 wk postpartum, the mares in this study were producing an average of 2.7% of BW in milk. 

Daily milk yield at 7 d postpartum was negatively related to the concentration of fiber in foal feces (r = –0.8144, *p* = 0.0138; Table 3). Mares producing more milk had foals with less fiber in their feces. From these data, it appears that the amount of milk available to the foals is a driving factor for the amount of fibrous feedstuffs the foals are consuming. There was a negative relationship between milk yield at 1 wk postpartum and the concentration of chromic oxide in foal feces (r = –0.6966, *p* = 0.0371; Table 3). It appears that foals from lower milk-producing mares coprophagized more than foals from high producing mares, thus increasing the fiber available in the GIT for the establishment of cellulolytic bacteria. The timeframe of approximately 7 d after birth appears to be critical in the development of the microbial community and may be influenced by changes in the foals’ diet, including milk yield and maternal feces.

There are a few limitations to consider with this study. The overall sample size was relatively small. The model used for detecting coprophagy in foals was novel; thus, estimates of variance were not available for calculating the sample size. The study set out to dose 12 mares with chromic oxide to detect coprophagy in their foals. However, three of those mares refused the marker, leaving nine mare and foal pairs that were sampled in the study. The findings of this study demonstrate that the sample size used was adequate for detecting differences. Despite the small sample size, the work presented here provides valuable knowledge to the field regarding the establishment of cellulolytic bacteria in foals and the important role of coprophagy.

## 4. Conclusions

Coprophagy is an important contributor to the establishment of fibrolytic bacteria in the foal’s GIT. Maternal feces appear to provide both fiber and live microbes to the foal. Thus, maternal feces may serve as both a prebiotic and a probiotic, supporting the developing microbial community in the foal’s GIT. As foals increase their coprophagic activity, more structural carbohydrates are available in the GIT, which results in more cellulolytic bacteria, particularly at 7 days of age. Furthermore, the amount of milk produced by the mare is an important factor influencing the coprophagy taking place and the subsequent fibrolytic bacteria in the foal. The first week of life is an important time of change in the GIT of foals regarding the establishment of important fiber-digesting bacteria, supported by the consumption of material feces.

## Figures and Tables

**Figure 1 animals-13-02718-f001:**
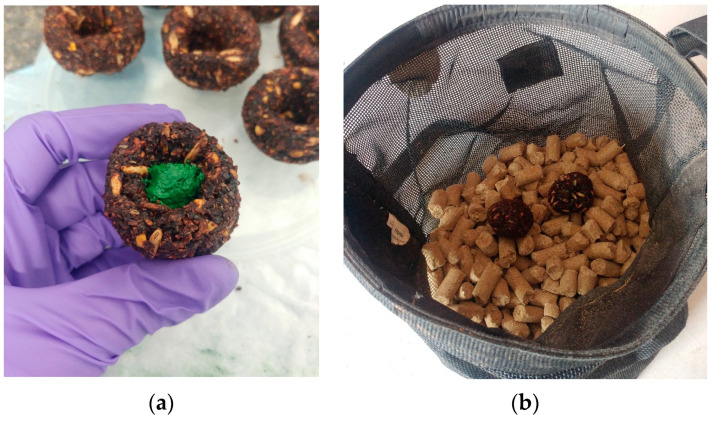
Chromic oxide mixed with molasses and placed in the center of a treat (**a**). Treats were fed to the mares in individual nosebags with concentrate meals (**b**).

**Figure 2 animals-13-02718-f002:**
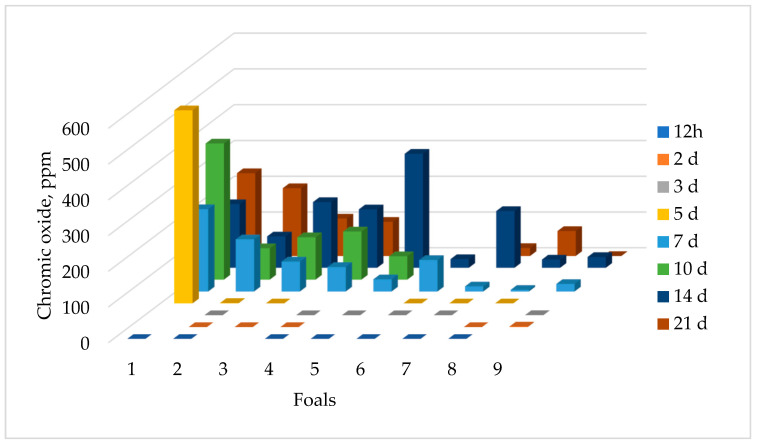
Concentration of chromic oxide in the feces of foals from mares fed 10 g chromic oxide per day. Each bar represents the concentration of chromic oxide in that foal’s feces on the specified sample day indicated by the color of the bar. Individual foals are represented by each row of bars moving toward the back of the figure.

**Figure 3 animals-13-02718-f003:**
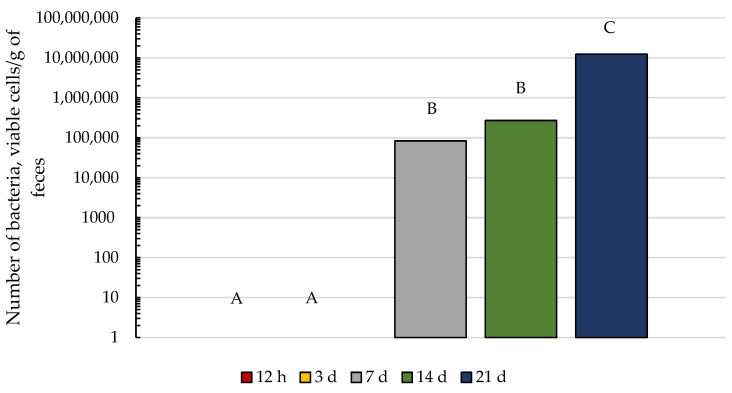
Changes over time in cellulolytic bacteria in foal feces. There was an increase in the number of cellulolytic bacteria in foal feces over time (*p* < 0.05). Data presented as back-transformed least squares means. Pooled SEM is Log_10_ transformed 0.3885. Letters A, B, and C represent differences between time points (*p* < 0.05).

**Figure 4 animals-13-02718-f004:**
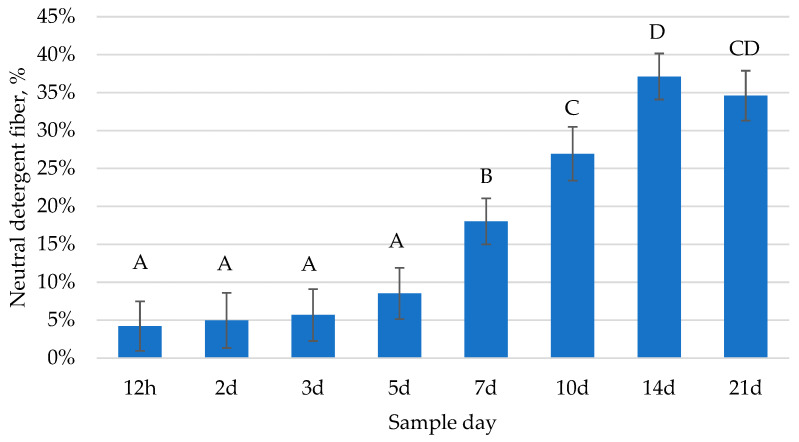
Changes over time in neutral detergent fiber (NDF) in foal feces from 12 h to 21 d of age (*p* < 0.0001). Data presented as least squares means ± SEM. Letters indicate differences between time points (*p* < 0.05).

**Table 1 animals-13-02718-t001:** Chemical composition of the pelleted concentrate and treats used for chromium dosing (DM basis).

Item	Concentrate	Treats
Chemical Composition ^a,b^		
DM, %	87.8	80.0
CP, %	16.2	12.6
NSC, % ^c^	39.1	48.6
Crude Fat, %	5.2	7.1
ADF, %	14.3	6.7
NDF, %	28.7	12.5
Starch, %	29.3	17.1
ESC, %	8.0	24.5
DE, Mcal/kg ^d^	1.44	1.69
Calcium, %	1.69	0.57
Phosphorus, %	1.06	0.55
Ca:P Ratio	1.59	1.04
Magnesium, %	0.43	0.42

^a^ Analyses performed by Dairy One (Ithaca, NY, USA) presented as means (n = 2); ^b^ Abbreviations: dry matter (DM), crude protein (CP), nonstructural carbohydrates (NSC), acid detergent fiber (ADF), neutral detergent fiber (NDF), ethanol soluble carbohydrates (ESC), digestible energy (DE). ^c^ NSC calculated by adding % starch and % water soluble carbohydrates; ^d^ DE estimated with the following equation [9]: DE (Mcal/kg) = 4.07 − 0.055 × (% ADF).

**Table 2 animals-13-02718-t002:** Excretion of chromic oxide in foal feces after oral dosing.

Time Post-Dose	Chromic Oxide, ppm	SEM
0–24 h	18.2 ^A^	55.0
25–48 h	1280.6 ^CD^	205.0
49–72 h	1926.7 ^D^	468.7
73–96 h	1236.3 ^C^	384.0
97–120 h	305.6 ^B^	101.8
***p*-value**	**<0.0001**	

Letters indicate differences between means (*p* < 0.05).

**Table 3 animals-13-02718-t003:** Relationships among milk yield, fecal fiber, chromic oxide, and fecal cellulolytic bacteria.

	n ^a^	r ^b^	*p*-Value
Fecal NDF, % vs. Chromic Oxide, ppm ^c^			
7 d	9	0.7878	0.0117
14 d	9	0.6648	0.0507
21 d	7	0.7526	0.0509
Fecal NDF, % vs. Cellulolytic bacteria, log_10_ cells/g feces			
7 d	9	0.8607	0.0029
14 d	9	0.0022	0.9955
21 d	7	−0.2273	0.6240
Chromic Oxide, ppm vs. Cellulolytic bacteria, log_10_ cells/g feces			
7 d	9	0.9703	<0.0001
14 d	9	−0.4455	0.2294
21 d	7	0.3683	0.4163
Fecal NDF, % vs. Milk Yield, L/d			
7 d	8	−0.8144	0.0138
14 d	8	0.1398	0.7413
Chromic Oxide, ppm vs. Milk yield, L/d			
7 d	8	–0.6966	0.0371
14 d	8	0.4805	0.2281

^a^ Number of foal fecal samples; ^b^ Pearson’s correlation coefficient; ^c^ NDF: Neutral detergent fiber.

**Table 4 animals-13-02718-t004:** Milk yield in L/d at 7 and 14 d post-foaling.

Horse	7 d	14 d
1	8.22	N/A ^a^
2	12.96	15.42
3	11.76	13.68
4	12.93	14.34
5	11.52	13.02
6	13.26	17.82
7	14.04	19.68
8	8.78	11.10
9	17.39	19.92
Mean	12.33	15.62
SEM	0.99	1.06
*p*-value	0.0386	

^a^ N/A: milk yield was not estimated in these mare.

## Data Availability

The raw data presented in this paper are available from the authors on request.

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
