# Peer review of "The Establishment of Fibrolytic Bacteria in the Foal Gastrointestinal Tract Is Related to the Occurrence of Coprophagy by Foals"

_animals, 2023, doi:10.3390/ani13172718_

Round 1

Reviewer 1 Report

A really nice study answering questions on both coprophagy and early fibre digestion. These data support other studies that show feeding a fibre based diet to foals provides adequate nutrition for growth compared to concentrates and your data help explain this. 

Given that its quite a short paper for the readership the methods for caclulating marker recovery and the lab work could have a little more detail rather than just citing the previous studies.

My other main comment is while the limitations are very subtly highlighted throughout before concluding the work would benefit from highlighting the limitations of the work to help the reader place the data into context. E.g. the overall sample size was small and then you could only collect samples from a sub sample of animals. Clearly the findings show the size of the sample was not an issue to detect differences but drawing conclusions on a small cohort has limitations. If you highlighted the limitation but identify not withstanding these that you believe the work has valuable input then you truly acknowledge the small sample but its valuable data to the field.

Reviewer 2 Report

It was a very clear, nice, and interesting article to read; please consider the following edits. Thank you

Please re-check and cite all the statements from the previously published literature throughout the manuscript.

Line 2: Please write “The” instead of “the”.

Line 2-3: I suggest minor changes in your title. So, It will read as,

The Establishment of Fibrolytic Bacteria in the Foal Gastrointestinal Tract is Related to the Occurrence of Coprophagy by Foals

Line 71,87,112,130,144,157,180,195,224,253,302: Typesetter, please check the formatting.

Line 194: Dear Authors, please write your results and discussion separately. Results should only contain your work and should not contain any citations or references. The discussion section should be separate from the results.

Table 1: Please check the formatting of Table one. Line numbers are on the left corner of the table and are confused with the values of the experiments.

Table 2: Same as suggested for Table 1.

Tables 3 and 4 are fine.
